# Population genomics of Sitka black-tailed deer supports invasive species management and ecological restoration on islands

Brock T. Burgess [1], Robyn L. Irvine [2] & Michael A. Russello [1✉]

Invasive mammals represent a critical threat to island biodiversity; eradications can result in ecological restoration yet may fail in the absence of key population parameters. Over-browsing by invasive Sitka black-tailed deer (*Odocoileus hemionus sitkensis*) is causing severe ecological and cultural impacts across the Haida Gwaii archipelago (Canada). Previous eradication attempts demonstrate forest regeneration upon deer removal, but reinvasion reverses conservation gains. Here we use restriction-site associated DNA sequencing (12,947 SNPs) to investigate connectivity and gene flow of invasive deer ($n = 181$) across 15 islands, revealing little structure throughout Haida Gwaii and identifying the large, central island of Moresby (>2600 km²) as the greatest source of migrants. As a result, the archipelago itself should be considered the primary eradication unit, with the exception of geographically isolated islands like SGang Gwaay. Thus, limiting eradications to isolated islands combined with controlled culling and enhanced biosecurity may be the most effective strategies for achieving ecological restoration goals.

¹ Department of Biology, The University of British Columbia, Kelowna, BC, Canada. ² Protected Areas Establishment and Conservation Directorate, Parks Canada Agency, Gatineau, QC, Canada. ✉email: michael.russello@ubc.ca

nvasive mammalian species are the primary cause of animal extinction on islands and represent one of the most significant threats to insular biodiversity[1]. The control and eradication of invasive mammals have become increasingly common tools used on islands globally[2], leading to substantial conservation gains in the form of ecological restoration and endemic species recovery[1,3]. Yet, efforts to remove such species are costly and subject to failure; for example, mammal eradication attempts on islands throughout the world have resulted in a ~15% failure rate[4]. Eradication failures, particularly those resulting from rapid recolonization, are often due to knowledge gaps of key population parameters, including population connectivity and dispersal capacity[5]. As such, there is a need for quantitative tools to assist managers in the decision-making process before, during, and after management operations to increase chances of successful ecological restoration, while also minimizing the costs associated with eradication failure.

Genetic and genomic tools have a proven history and promising future for informing invasive mammal management[6,7]. An early application involved population genetic analysis of invasive Norway rats (Rattus norvegicus) across 18 islands representing five archipelagos off the coast of France, where microsatellite genotypic data were used to determine eradication units (i.e., islands with sufficient gene flow to be considered single populations) and highlight the importance of conducting pre-eradication genetic surveys to minimize eradication failure[8]. Since then, using genetic data for defining eradication units has become an effective strategy for invasive species management on islands, evidenced by its many applications and recommendations[5,9,10]. Genetic data can also be used to infer invasion pathways, including the source(s) of invasion, extent and direction of migration, and the dispersal capacity of a particular species[7,11]. Moreover, genotyping individuals before and after an eradication can better inform management outcomes by identifying the source of reinvasion and guiding subsequent planning[12–14].

Sitka black-tailed deer (Odocoileus hemionus sitkensis) were introduced to Haida Gwaii, an archipelago off the coast of British Columbia (Canada), in the late 19th century and have become a widespread invasive species in the 50–80 generations since[15]. Haida Gwaii is rich in biodiversity, containing more unique endemic subspecies than any other equal-sized area in Canada[16]. Deer over-browsing has had tremendous negative consequences within this system[15]; islands with deer populations accommodate less plant[17], insect[18], and songbird[19] species diversity, smaller population sizes, and reduced belowground nutrient cycling[20] when compared to deer-free islands. The Indigenous culture of the Haida Nation is also affected, as their fundamental knowledge that all things are interconnected (Yahgudaang) is disrupted by the negative effects of invasive species. The Haida also integrate into their cultural practices many species that are the preferred browse species of deer; for example, Ts'uu (western redcedar; Thuya plicata) is used for many purposes including infrastructure and art, while Ts'iihlinjaaw (devil's club; Oplopanax horridus) is highly valued for its spiritual and medicinal properties[21,22]. There is compelling evidence of forest regeneration and ecosystem recovery upon removal or reduction of deer from invaded islands[23], yet recent eradication attempts targeting islands in the Gwaii Haanas National Park Reserve, National Marine Conservation Area, and Haida Heritage Site in Haida Gwaii (hereafter referred to as Gwaii Haanas) remain incomplete[2].

In response to the ongoing reduction in vegetation and deteriorating habitat quality for several unique ecologically and culturally important species in Gwaii Haanas, the Llgaay gwii sdiihlda (Restoring Balance) Conservation and Restoration project was initiated in 2014. The central aim of this project was to restore forest ecosystems by removing invasive deer from six islands in Juan Perez Sound, which was targeted for active

conservation due to its high ecological and cultural value. A seventh island and UNESCO World Heritage Site, SGang Gwaay, was also targeted for deer management. Deer have been culled on SGang Gwaay between 1998–2003 and again in 2018, with the population being reduced to <10 individuals by the end of the earlier management operation[15]. Unlike the islands in Juan Perez Sound, SGang Gwaay is located at the southern end of Gwaii Haanas, roughly 3 km from its nearest neighbor (Moresby), making it the most geographically isolated island in this study. The project made use of various hunting methods employing firearms, which were challenging given the: (1) height of trees that have forced aerial eradication efforts to occur at more difficult, higher elevations; and (2) presence of large areas of blowdown in which deer can hide, rendering pursuit difficult or impossible. In addition, Sitka black-tailed deer are able swimmers, requiring boat support during population reductions and necessitating biosecurity plans to account for immigration and island connectivity. Given the incomplete eradication associated with the Llgaay gwii sdiihlda project and the knowledge gained from prior restoration work, Parks Canada and the Council of the Haida Nation determined that additional information on deer movement and gene flow was required to delineate areas that could be maintained as deer-free.

To provide actionable information to guide invasive deer management and ecological restoration within Gwaii Haanas, here we employ restriction-site associated DNA sequencing (RADseq) to (1) reconstruct the distribution of Sitka black-tailed deer genetic variation within and among islands across the archipelago; and (2) quantify the direction, magnitude, and sex-bias of gene flow. Population genomic analyses using 12,947 neutral single nucleotide polymorphisms (SNPs) revealed a high degree of connectivity among 181 deer across 15 islands, suggesting there are minimal barriers to deer movement throughout the system. SGang Gwaay is a notable exception in this regard, as deer on this island displayed a signature of genetic differentiation relative to the rest of the archipelago, likely influenced by its history of culling. Contemporary migration analysis further indicated that Moresby, the largest island within Gwaii Haanas, is the greatest source of migrants and driver of gene flow to neighboring islands, consistent with island biogeography theory[24]. These findings have important implications for managing invasive deer and achieving ecological restoration, as they suggest that except for a few, isolated islands, the archipelago itself is the primary eradication unit. Thus, efforts to remove deer from targeted islands within Gwaii Haanas, specifically, and Haida Gwaii, more broadly, will likely result in reinvasion unless additional biosecurity measures, such as fencing, are implemented.

## Results

**Genotyping and SNP filtering**. Sequencing of four RADseq libraries yielded 1,013,204,998 total forward and reverse DNA sequence reads from Sitka black-tailed deer harvested across 15 islands in Haida Gwaii (Fig. 1 and Supplementary Table 1). Final genotypic filtering parameters were selected using STACKS[25] ($R = 0.9$, $min\_maf = 0.03$) and a single individual was removed due to low coverage (<5x), leaving 181 unique individuals genotyped at 12,961 loci (Supplementary Table 2). We detected and removed 12 outlier loci using BayeScan[26], and an additional two that deviated from Hardy–Weinberg equilibrium, resulting in 12,947 neutral loci for subsequent population genomic analyses. Genotyping error was 4.20% within libraries and 3.34% among libraries (Supplementary Table 2).

**Genetic diversity and population structure**. Genetic diversity metrics were calculated for all putative populations (i.e., islands)

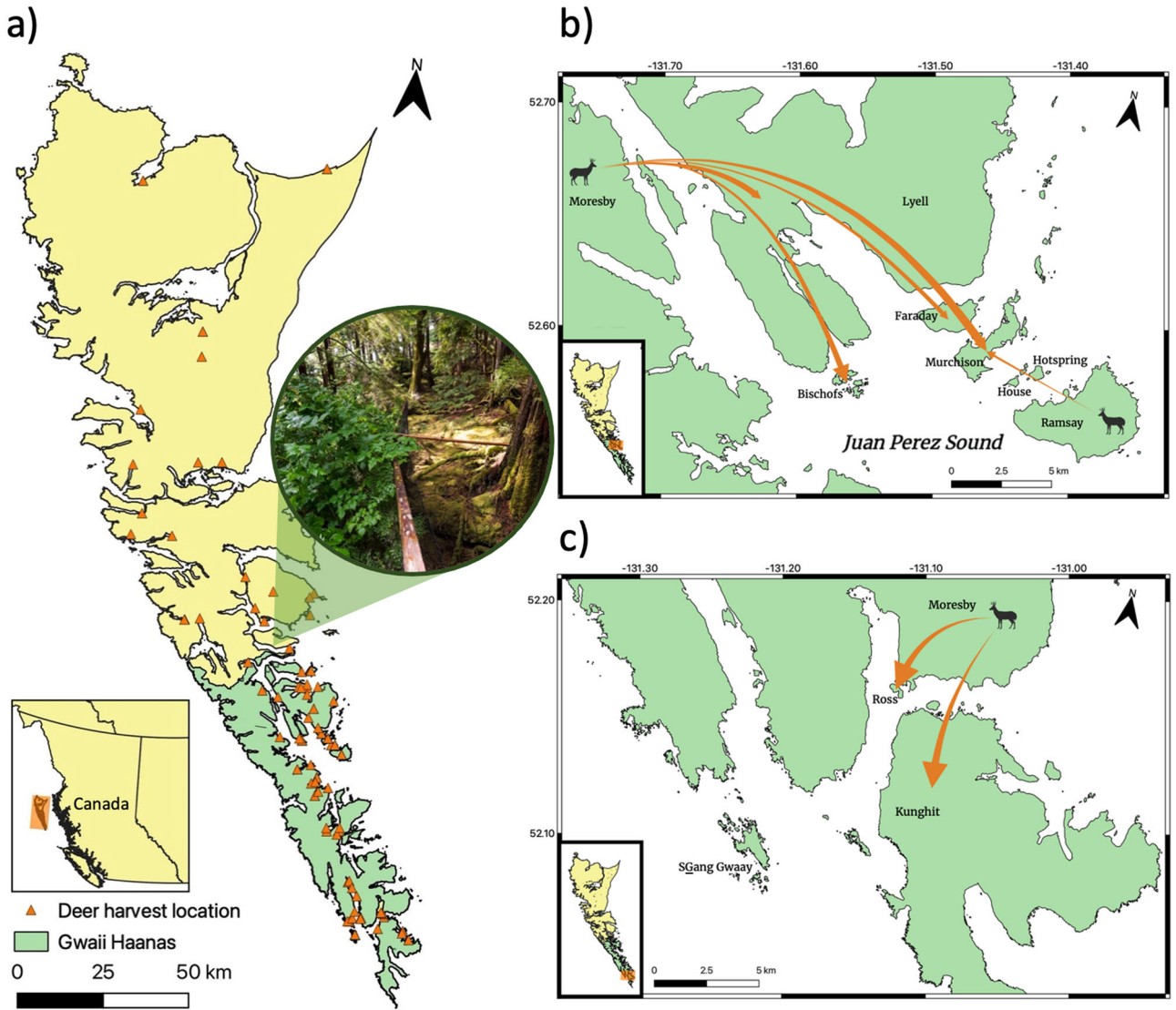

**Fig. 1 Sampling distribution of Sitka black-tailed deer in Haida Gwaii and directional migration within Gwaii Haanas.** Haida Gwaii is a densely forested archipelago ~60–100 km off the western coast of Canada and is threatened by invasive Sitka black-tailed deer. **a** Deer were harvested across the archipelago, including Gwaii Haanas (green), and harvest locations representing the 181 individuals genotyped for this study are shown (orange triangles). The image (© Parks Canada) depicts a deer exclosure established on Kunga Island. **b** Contemporary migration of deer into northern Juan Perez Sound from Moresby Island, as well as the weaker signal of migration coming from Ramsay Island. Arrow width is relative to directional migration rates, estimated as the number of migrants per generation. **c** Contemporary migration of deer to Ross and Kunghit Islands in the south of Gwaii Haanas. Note that the two other instances of significant migration indicated in the text and Supplementary Table 5 are outside of Gwaii Haanas and are not depicted here. Locator maps are set in the bottom left of each panel, with the enlarged areas highlighted in orange.

where $n \geq 2$ (Table 1). Observed heterozygosity varied across the archipelago, ranging from 0.137 (SGang Gwaay) to 0.248 (Hotspring), and were lower than expected across all islands except for House (Table 1); these patterns were mirrored in the unbiased measures of heterozygosity calculated using all sites, with SGang Gwaay exhibiting approximately half the levels (0.0003) of the highest values detected elsewhere (0.00006; Hotspring, Ramsay, Ross) (Table 1). High levels of inbreeding were detected for several islands and ranged from −0.030 (House) to 0.198 (Louise). Including all islands, the overall observed heterozygosity was 0.215 (95% CI: 0.213–0.217) and the overall inbreeding coefficient was significantly different than zero at 0.087 (95% CI: 0.084–0.089; $p < 0.001$).

A principal component analysis (PCA) using all islands revealed one clearly distinct cluster in the parameter space corresponding to

the individuals from SGang Gwaay (Fig. 2a), with most of the remaining individuals clustering closely together. We removed SGang Gwaay to further investigate the presence of discrete genetic units among the remaining islands. In this analysis, most individuals from Ramsay clustered together separately from, but close to, the rest of the individuals sampled archipelago-wide (Fig. 2b).

Bayesian clustering analyses implemented in STRUCTURE[27] revealed $K = 2$ as the optimal number of genetic clusters (Supplementary Table 3), with deer from SGang Gwaay being distinct relative to the rest of the archipelago (Fig. 2c). The remaining individuals largely comprised a single cluster, with some evidence of structure within northern Juan Perez Sound when the analysis was repeated in the absence of SGang Gwaay (Figs. 1b, 2d and Supplementary Table 4).

**Table 1 Genetic diversity by island for Sitka black-tailed deer.**

| Island | N | $N_A$ | $H_O$ | $H_E$ | $G_{IS}$ | $uH_O$ | $uH_E$ |
|---|---|---|---|---|---|---|---|
| Bischofs | 7 | 1.743 | 0.216 | 0.244 | 0.114 | 0.00005 | 0.00005 |
| Burnaby | 2 | 1.422 | 0.202 | 0.245 | 0.174 | 0.00004 | 0.00005 |
| Faraday | 8 | 1.751 | 0.220 | 0.242 | 0.093 | 0.00005 | 0.00005 |
| Graham | 11 | 1.839 | 0.230 | 0.249 | 0.078 | 0.00005 | 0.00006 |
| Hotspring | 2 | 1.463 | 0.248 | 0.254 | 0.022 | 0.00006 | 0.00004 |
| House | 7 | 1.613 | 0.226 | 0.220 | −0.030 | 0.00005 | 0.00005 |
| Kunghit | 5 | 1.623 | 0.199 | 0.234 | 0.147 | 0.00005 | 0.00005 |
| Louise | 7 | 1.706 | 0.192 | 0.239 | 0.198 | 0.00005 | 0.00005 |
| Lyell | 16 | 1.882 | 0.195 | 0.236 | 0.173 | 0.00005 | 0.00005 |
| Moresby | 37 | 1.970 | 0.215 | 0.243 | 0.115 | 0.00005 | 0.00006 |
| Murchison | 23 | 1.936 | 0.223 | 0.248 | 0.102 | 0.00005 | 0.00006 |
| Ramsay | 28 | 1.946 | 0.239 | 0.244 | 0.020 | 0.00006 | 0.00006 |
| Ross | 4 | 1.658 | 0.245 | 0.259 | 0.054 | 0.00006 | 0.00005 |
| SGang Gwaay | 23 | 1.454 | 0.137 | 0.141 | 0.024 | 0.00003 | 0.00003 |
| Tanu | 1 | --- | --- | --- | --- | --- | --- |

$N_A$ average number of alleles, $H_O$ observed heterozygosity, $H_E$ expected heterozygosity, $G_{IS}$ inbreeding coefficient, $uH_O$ unbiased observed heterozygosity, $uH_E$ unbiased expected heterozygosity.

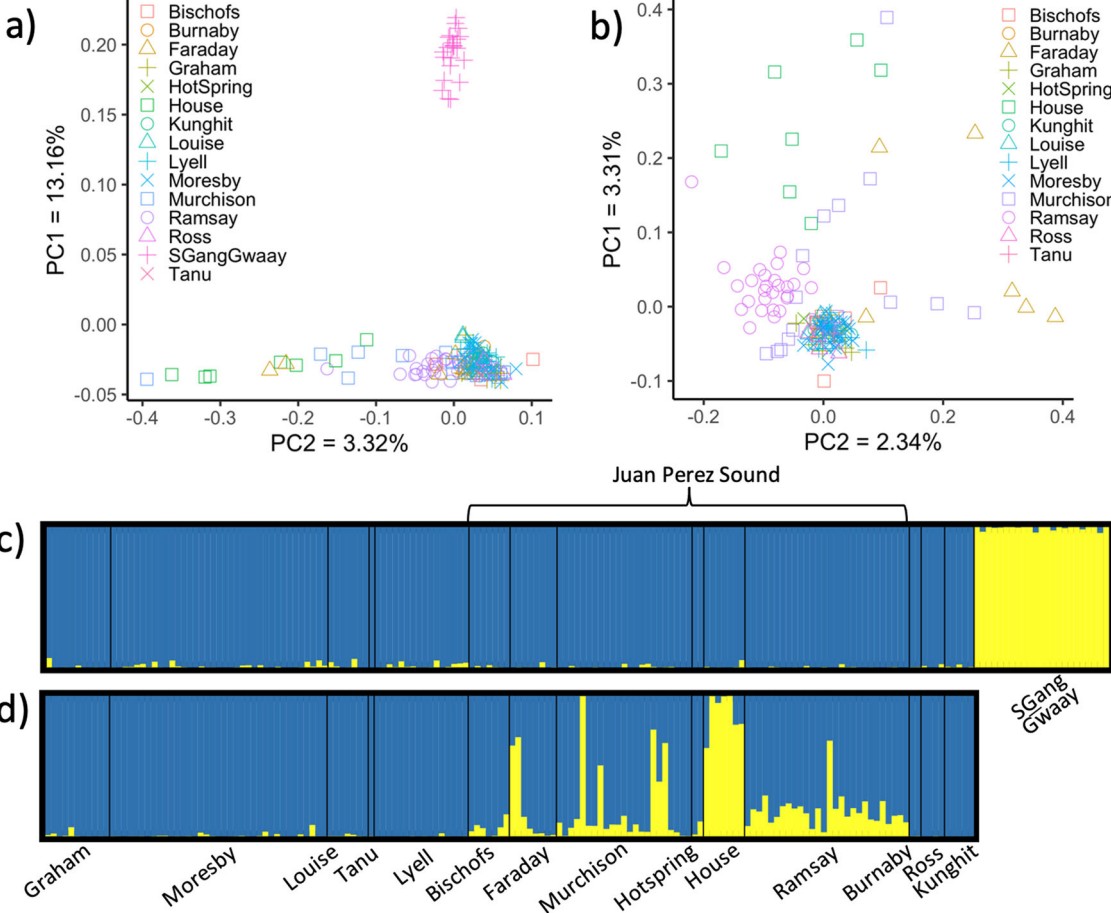

**Fig. 2 Genetic clustering and population structure of Sitka black-tailed deer in Haida Gwaii.** All analyses were performed using Sitka black-tailed deer genotyped at 12,947 neutral SNPs. **a** Principal component analysis using all deer genotyped in the study (n = 181). **b** Same as (**a**), but with deer from SGang Gwaay removed (n = 158). **c** Bayesian clustering using all deer showing the proportion of ancestry of each individual (vertical bar) assigned to inferred populations (K = 2) that are designated by color. Harvest locations for each deer are listed underneath, structured by island. **d** same as (**c**), but with deer from SGang Gwaay removed.

An analysis of molecular variance (AMOVA)[28] indicated that most of the genetic variation within Sitka black-tailed deer in Haida Gwaii was contained within islands (92.5%; p < 0.001), but with a significant amount also found among islands (7.50%; p < 0.001). When SGang Gwaay was removed and the analysis repeated, the vast majority of genetic variation was contained within (98.2%; p < 0.001) rather than among islands (1.80%; p < 0.001). Tests of pairwise genetic differentiation between all islands where n ≥ 2 were consistent with the PCA, STRUCTURE, and AMOVA (Table 2).

**Table 2 Pairwise genetic differentiation for Sitka black-tailed deer between islands, shown below the diagonal.**

| | Bischofs | Faraday | Graham | House | Kunghit | Louise | Lyell | Moresby | Murchison | Ramsay | Ross | SGang Gwaay |
|---|---|---|---|---|---|---|---|---|---|---|---|---|
| Bischofs | -- | * | ** | ** | * | * | * | ** | NS | ** | NS | ** |
| Faraday | 0.022 | -- | ** | ** | ** | ** | ** | ** | * | ** | * | ** |
| Graham | 0.016 | 0.029 | -- | ** | NS | * | ** | ** | ** | ** | NS | ** |
| House | 0.075 | 0.063 | 0.070 | -- | ** | ** | ** | ** | ** | ** | ** | ** |
| Kunghit | 0.013 | 0.029 | 0.013 | 0.080 | -- | * | * | * | * | ** | NS | ** |
| Louise | 0.009 | 0.026 | 0.010 | 0.071 | 0.007 | -- | * | NS | * | ** | NS | ** |
| Lyell | 0.008 | 0.020 | 0.011 | 0.066 | 0.010 | 0.005 | -- | ** | ** | ** | NS | ** |
| Moresby | 0.009 | 0.023 | 0.009 | 0.060 | 0.005 | 0.003 | 0.006 | -- | ** | ** | NS | ** |
| Murchison | 0.008 | 0.018 | 0.018 | 0.047 | 0.015 | 0.012 | 0.013 | 0.012 | -- | ** | NS | ** |
| Ramsay | 0.017 | 0.033 | 0.020 | 0.050 | 0.022 | 0.017 | 0.017 | 0.013 | 0.016 | -- | * | ** |
| Ross | 0.008 | 0.026 | 0.009 | 0.074 | 0.003 | −0.002 | 0.001 | 0.003 | 0.010 | 0.016 | -- | ** |
| SGang Gwaay | 0.280 | 0.283 | 0.262 | 0.319 | 0.290 | 0.282 | 0.246 | 0.220 | 0.236 | 0.232 | 0.295 | -- |

Significance is indicated above the diagonal.
NS non-significant.
*$p < 0.05$, **$p < 0.01$.

**Migration, kinship, and sex-biases**. Contemporary migration rates between all islands were estimated using the Bayesian approach implemented in BA3-SNPs[29,30], revealing significant gene flow between nine island pairs (Fig. 1b–c and Supplementary Table 5). Moresby was the greatest source of migrants; significant migration was detected from Moresby to Bischofs, Faraday, Graham, Kunghit, Louise, Lyell, Murchison, and Ross Islands. Of these, migration rates, measured as the estimated number of migrants per generation, ranged from 0.058 (Moresby to Faraday) to 0.125 (Moresby to Graham). In addition, significant migration was detected from Ramsay to Murchison ($m = 0.031$).

Of the 16,290 pairwise kinship coefficients[31] calculated, 280 represented first-order relatives with values ≥0.20; however, most of these estimates ($n = 255$) were between individuals harvested from SGang Gwaay. In that respect, all kinship coefficients between unique pairs of deer harvested from SGang Gwaay were extremely high, ranging from 0.23 to 0.47 and averaging 0.32. Most of the remaining 25 pairs were also between individuals harvested from the same island, but nine pairs of first-order relatives were detected between deer harvested from different islands, all of which were within northern Juan Perez Sound (Murchison, House, Ramsay, Faraday, Bischofs; Fig. 3). An additional 39 pairs exhibited kinship coefficients between 0.10–0.20 and were second-order relatives, 16 of which were between individuals harvested from different islands in northern Juan Perez Sound (Fig. 3).

Using a pairwise differentiation ($\theta$)[32] approach for detecting sex-biased dispersal in Sitka black-tailed deer in Haida Gwaii, we found that females ($\theta = 0.102$) were significantly more differentiated than males ($\theta = 0.055$; $p = 0.040$; two sample $t$-test). Across all islands, males scored lower mean (mAIC) and variance (vAIC) of population assignment indices[33] ($-27.7 \pm 32.4$ SE and 62390.4, respectively) compared to females ($42.1 \pm 36.8$ SE and 73461.3, respectively). Neither the difference in means ($p = 0.16$; Mann–Whitney $U$-test) or variance ($p = 0.56$; $F$-test) were significant.

## Discussion

Islands are home to many of the world's most unique and at-risk species; however, these systems are especially vulnerable to the negative impacts of introduced species, particularly mammals[3,34]. As a result, management interventions such as invasive mammal eradication or control have become important strategies for mitigating biodiversity losses[1,2]. While proven to be effective when implemented successfully, eradications are often subject to failure

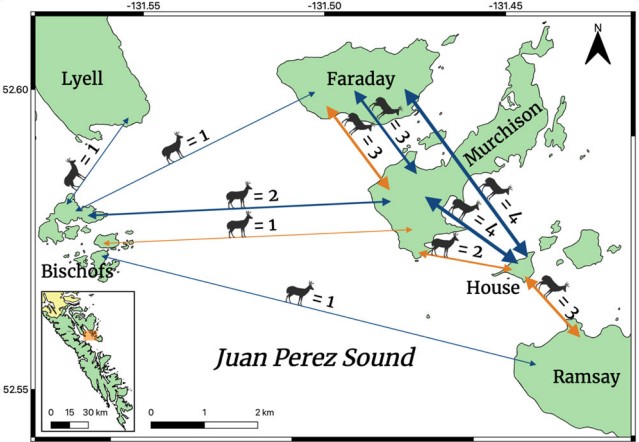

**Fig. 3 Inter-island close relatives identified between Sitka black-tailed deer in Gwaii Haanas.** The pairwise kinship analysis was performed using all 181 Sitka black-tailed deer in this study; shown is every detected inter-island first (orange; $n = 9$) or second (blue; $n = 16$) order relative pair, each within Juan Perez Sound. The number of related deer pairs found between islands is shown above each line, which are weighted accordingly. A locator map is set in the bottom left corner, with the enlarged area highlighted in orange.

due to the complex nature of species invasions; chances of success can be greatly improved with a priori knowledge of their invasion history, including population connectivity, contemporary movement patterns, and dispersal capacity[7]. Knowledge of these processes can further inform the development of biosecurity plans to maintain conservation gains from eradication or control programs over time. Population genomics provides a framework for filling these knowledge gaps and reducing the probability of eradication failure. Here we demonstrate the application of population genomics to invasive Sitka black-tailed deer management in Haida Gwaii (Canada), an ecologically diverse and culturally important system with unique biodiversity and essential ecosystem processes under threat[17–20].

Our results revealed a remarkable lack of population genetic structure for Sitka black-tailed deer among sampled islands in Haida Gwaii, suggesting a high degree of connectivity across the system. These findings were consistently supported by a range of analyses, including a general lack of clustering in the PCAs and STRUCTURE analysis, and low to absent levels of differentiation revealed by AMOVA and pairwise $\theta$ values. Moreover, these

patterns of structure (or lack thereof) were reinforced by significant estimates of contemporary migration and direct detection of inter-island first- and second-order relatives. The exception to this overall trend was SGang Gwaay, which exhibited high genetic differentiation from the rest of the archipelago.

SGang Gwaay is situated at the southern end of Gwaii Haanas, roughly 3 km from its nearest neighbor (Moresby), rendering it more isolated than the other islands in our study. Yet, this distance is well within the known dispersal capacity of mule deer, generally[35,36], as well as demonstrated elsewhere within the archipelago. As a case in point, Ross and Kunghit, also located at the southern end of Gwaii Haanas, clustered more closely to distal islands, such as those in northern Juan Perez Sound (>50 km), than to SGang Gwaay (<10 km), despite large differences in geographic distance. However, the distinct pattern of differentiation observed in SGang Gwaay may not be due to a lack of connectivity per se, but more influenced by a history of population culling. The rapid reduction in population size between 1998–2003 may have led to the loss of genetic diversity in the SGang Gwaay population, reflected in the significantly lower levels of heterozygosity and higher levels of pairwise kinship observed here relative to the rest of the archipelago (Table 1). Pairwise kinship could also be artificially inflated due to the sex ratio of residual founders post-cull, as mule deer have a propensity to display polygynous mating behavior[37]. For example, following the 1998–2003 cull, the sex ratio of deer on SGang Gwaay was reported to be heavily biased towards females (3:1)[15]. If this pattern continued, it is possible that the contemporary population on SGang Gwaay may be the progeny of a very small number of males, which could contribute to the high degree of differentiation seen here. However, given that genetic differentiation is a function of both connectivity and population size, we cannot exclude the possibility that the apparent lack of gene flow detected between SGang Gwaay and other islands is due to geographic isolation rather than a previous population bottleneck.

On a broader level, contemporary migration analysis revealed Moresby as the greatest source of migrants to northern Juan Perez Sound as well as islands to the south. This result was not surprising; island biogeography theory predicts that larger, more proximal islands will be greater suppliers of migrants than smaller, distal islands[24]. After Graham, Moresby is the largest island in Haida Gwaii, spanning >2600 km², almost half of which is within Gwaii Haanas. Outside of the northern Juan Perez Sound island cluster, all of the islands included in this study are located <2.5 km from Moresby, well within the distance over which deer have been observed swimming according to multiple lines of anecdotal evidence[38]. Across northern Juan Perez Sound, the islands form a natural stepping stone from Moresby to Ramsay, with <2 km separating each island in the chain. In that regard, we found significant evidence of gene flow from Moresby to all but the most distal islands in northern Juan Perez Sound: House, Hotspring, and Ramsay. Yet, our analysis did reveal significant gene flow from Ramsay to Murchison, suggesting deer are indeed moving among these islands. Considering the small area of House and Hotspring (0.33 km² and 0.17 km², respectively), and a low number of individuals harvested despite exhaustive hunting ($n = 7$ and $n = 2$, respectively), these islands likely do not support self-sustaining deer populations and rely on immigration directly from Ramsay and indirectly from Moresby. Our pairwise kinship analysis provides some support for this hypothesis, as we detected four first-order relatives between individuals sampled on House and those from Ramsay and Murchison (Fig. 3). Likewise, eight second-order relatives were identified between individuals sampled on House and either Murchison or Faraday (Fig. 3). Taken together, these findings of high pairwise kinship between House individuals and

neighboring islands suggest that deer on House may be comprised of recent immigrants from multiple sources. These inferences could further explain the notable genetic differentiation identified in the Bayesian clustering analysis; aside from SGang Gwaay, House was the only island that constituted a distinct cluster (Fig. 2d).

There was minimal evidence of sex-biased dispersal in Sitka black-tailed deer across Haida Gwaii, which is in contrast to the male-biased dispersal detected in multiple studies of mainland mule deer across different habitats and subspecies[35,36,39,40]. It is important to acknowledge, however, that tests for sex-biased dispersal assume comparisons are made between individuals from distinct populations; the overall lack of structure detected in this system led us to use the island of origin as the unit for comparison, which may have masked any signal if present. Moreover, the sample sizes per island used for these analyses were small, substantially reduced from the population-level estimates due to having to factor in the biology of the species (e.g., only using individuals aged ≥3 in accordance with observations of mule deer dispersal) and minimal sample size for an island to be included (e.g., $n \geq 2$ for both males and females).

Introduced deer have the potential to become impactful invasive species with negative consequences that extend beyond species and communities to entire ecosystems[41]. Consequently, many invasive deer populations are managed to maintain target population densities, which may take the form of seasonal hunting, serial culling, or eradication[2]. Deer can be profoundly difficult to eradicate due to the densely forested areas they often inhabit and the lack of permitted toxicants for large vertebrates in North America, leading to hunting being implemented as a common management strategy. In island systems, deer eradications have been successful across several systems and species globally[42–45]. An important consideration for island management is the ability of deer to swim, as eradication strategies must be designed to minimize the potential of reinvasion from nearby sources, which often requires the eradication of deer from multiple islands simultaneously (i.e., eradication unit).

For Sitka black-tailed deer management within Gwaii Haanas, our findings suggest that any future eradication efforts need to consider their high degree of connectivity across the archipelago. Since the archipelago itself is the primary eradication unit, efforts to eradicate individuals from targeted islands in the absence of a broader campaign would likely lead to reinvasion. This is especially true in northern Juan Perez Sound given the apparent connectivity and evidence of deer movement among islands. For example, as part of the Llgaay gwii sdiihlda project, deer were thought to have been completely removed from several of these islands in 2017-18, including Bischofs, House, Hotspring and Murchison, and the population on Ramsay was reduced to an estimated ten individuals. Based on our contemporary migration analysis, it is not surprising that these islands did not remain deer-free after eradication efforts due to the movement of individuals into Juan Perez Sound from the much larger source, Moresby. Given the lack of population genetic structure found within Moresby and its large supply of migrants to smaller, neighboring islands, any hopes of permanently removing deer from Gwaii Haanas would necessitate large-scale management operations inclusive of the deer residing on Moresby. To date, the largest island in which deer of any species have been successfully eradicated is Secretary Island (New Zealand; 81.4 km²), where red deer (*Cervus elaphus*) were completely removed in 2014[45]. The planned eradication to remove red deer from the much larger Resolution Island (New Zealand; 210 km²) is thus far incomplete, but it continues to remain an eradication target for the New Zealand Department of Conservation[46]. Considering the area of Moresby within Gwaii Haanas exceeds 1000 km², an eradication

operation of this scope and scale is well beyond the ability of current invasive species management approaches.

Instead of eradication, a more feasible alternative to managing deer in Gwaii Haanas may be the continued culling from critical conservation areas with the goal of reducing browse pressure enough to restore some of the ecological integrity of the system. Installation of deer fencing in combination with culling is another potential strategy; this approach has been used in New Zealand to maintain target deer densities within fenced areas that enable the achievement of ecological restoration targets. This strategy may be an option at the northern end of Gwaii Haanas, but would require extensive fencing to partition off zones; the maintenance of such fencing in a coastal forest with high stem density and frequent blowdown events, however, would constitute a significant ongoing cost. In addition, management would need to establish methods for determining target deer densities that would maximize restoration efforts on islands of high conservation and cultural value. These methods would be best informed by studies that compare the response of vegetation and other taxa on islands with deer numbers reduced to those on islands with deer removed completely, ultimately allowing the future benefit of large-scale eradications to be effectively contextualized.

The only exception to the widespread connectivity of deer throughout Gwaii Haanas within this sample set is SGang Gwaay (1.35 km²), where a standalone eradication operation could potentially be successful and durable. For this study, deer were selectively harvested from the southern end of Gwaii Haanas, including twelve deer from the southern coast of Moresby, to best evaluate any possible connectivity with deer on SGang Gwaay. Given the heavy sampling that took place on SGang Gwaay ($n = 23$), we expected that any gene flow to or from Moresby would be detected. Although the high level of genetic differentiation observed in deer on SGang Gwaay was possibly inflated following the substantial population reduction in 2001, the island may have been founded by a small number of individuals followed by limited to absent immigration. Our results appear to be consistent with this idea, given the lack of gene flow detected in either direction between SGang Gwaay and neighboring islands, suggesting the eradication of deer here may be a promising management target in Gwaii Haanas. SGang Gwaay is also recognized as a UNESCO World Heritage Site and previous deer management has led to effective, but temporary, recovery of the coastal hypermaritime forest ecosystem. Collectively, these results and observations provide support for SGang Gwaay as a viable short-term and moderately sized option on which to pursue more comprehensive restoration efforts.

Overall, our study highlights the value of conducting genetic surveys as part of invasive species management programs in island systems. In Haida Gwaii, we found that invasive Sitka black-tailed deer exhibited substantial connectivity across the system, despite being spread over many different islands. Additionally, we found compelling evidence of contemporary deer movement among closely situated islands, with the largest and central island of Moresby being the greatest source of migrants; as a result, eradicating deer from any one island within our dataset (with the exception of SGang Gwaay) would likely result in reinvasion. The importance of these findings cannot be overstated; failed eradications typically lead to invasive populations returning to pre-eradication densities, thereby rendering any conservation gains temporarily. Failures further lead to additional financial costs to complete the eradication, while also incurring future opportunity costs that delay other interventions necessary to achieve restoration goals[47]. Taken together, these findings will inform future management strategies involving Sitka black-tailed deer within Gwaii Haanas and across all of Haida Gwaii as efforts continue to restore the ecological integrity of the system.

## Methods

**Study sites and sample collection.** Sitka black-tailed deer ear or muscle tissue samples were collected by Parks Canada staff and contractors, as well as from an island-wide hunter sample donation program that was run through the local cutting room. Animals taken within Gwaii Haanas were harvested under a Parks Canada Agency Animal Care Permit between 2017–2019 and those donated by hunters were animals taken under provincial hunting licenses or First Nations hunting rights; all relevant ethical regulations for animal research were followed. Throughout the project, several approaches were used to harvest deer, including bait station hunting, shoreline hunting, detection dog hunting, and aerial hunting. Ear or muscle tissue was removed from each individual in the field after death and was either assigned to the island of residence or geographic coordinates (where recorded). Ear tissue was placed in 95% ethanol immediately after collection in the field. In addition, teeth were collected and sent to Matson's Laboratory (USA) for age assessment. The final sample inventory for population genomic analysis included 182 individuals from 13 islands within Gwaii Haanas, and two islands outside of the national park (Fig. 1 and Supplementary Table 1).

**RADseq library designs and preparation.** Genomic DNA was extracted from ear tissue from 182 individuals using a Qiagen® DNeasy® Blood and Tissue Kit following the manufacturer's suggested protocol. Following standardization of DNA quantity (1000 ng), we constructed four RADseq libraries using a SbfI RADseq protocol[48,49]. Replicate samples were included within ($n = 5$) and among ($n = 4$) libraries to later estimate genotyping error as the discordance of genotype calls between replicate samples using a custom python script (https://github.com/bsjodin/genoerrorcalc). Samples within each library were pooled together and sheared into fragments of ~500 base pairs (bp) using sonication (Bioruptor® NGS; Diagenode). A targeted size selection instrument (Pippen Prep™; Sage Science) was used to isolate DNA fragments between 350 and 600 bp long. Isolated fragments were amplified via PCR using Phusion PCR reagents (New England Biolabs® Inc.) and barcode-specific primers prior to further size selection targeting fragments between 350 and 650 bp long. The resulting libraries were sequenced using one full lane each (four lanes total) of paired-end 125 bp sequencing on either an Illumina HiSeq2500 or HiSeq4000 at the McGill University and Génome Québec Innovation Centre.

**Data processing and SNP genotyping.** Raw sequencing data for each library were processed using the STACKS v.2.0 bioinformatics workflow[25] to demultiplex and clean sequence reads (i.e., remove RAD barcodes, PCR duplicates, and poor quality bases). Reads were trimmed to 100 bp, as base-call accuracy decreased by an order of magnitude between 100 and 125 bp. Processed and filtered reads were interleaved and aligned to the *O. h. hemionus* reference genome (GenBank assembly accession: GCA_004115125.1) using the *bwa mem* algorithm in BWA[50]. After alignment, the *ref_map* module was implemented to catalog and call loci for subsequent population genomic analyses. Throughout, important checks for quality assessment (i.e., mean percentage of aligned reads >50%, mean percentage of removed reads due to soft clipping <4%, mean per locus coverage >10x) were followed as recommended[51].

The STACKS *populations* module was used along with VCFtools v.0.1.16[52] to determine optimal filtering parameters via a sensitivity analysis (Supplementary Table 2). This involved testing six values of *R*, the minimum proportion of individuals that must contain a locus for it to be called. For each value of *R*, five different minimum minor allele frequencies (*min_maf*) were tested, resulting in 30 total runs using *populations*. Throughout each combination of *R* and *min_maf*, we evaluated the quantity of retained individuals and loci by comparing them to the amount of missing data per individual and the average depth of coverage per locus. The final parameters were selected to maximize the number of loci while maintaining low missing data, a high mean depth of coverage, and reasonable within/among genotyping error rates (Supplementary Table 2). Each *population* run was further filtered to ensure all retained individuals had a mean depth of coverage >5x and a maximum observed heterozygosity = 0.5. All individuals were filtered to only retain a single SNP per locus, reducing the potential for linkage disequilibrium; low coverage individuals (<5x) were removed.

Detection of outlier loci[53] was performed with BayeScan v.2.1[26] using 100,000 iterations with a 50,000 iteration burn-in period and prior odds of 100. Loci were considered outliers and removed if they had a mean *q*-value <0.20 averaged over five runs. Following outlier removal, we removed loci that significantly ($\alpha = 0.05$) deviated from Hardy–Weinberg equilibrium in >50% of the populations using VCFtools v.0.1.16[52]. Where necessary, data were converted from vcf to appropriate input file formats using PGDSpider v.2.1.1.5[54].

**Genetic diversity and population structure.** Using our putatively neutral dataset, standard measures of genetic diversity were calculated for putative populations structured by island using GenoDive v.3.0[55], including the observed and effective numbers of alleles, observed and expected heterozygosities, and levels of inbreeding ($G_{IS}$).

To evaluate population connectivity and determine eradication units, we first conducted PCA using the *SNPRelate* R package[56], which were then visualized by employing the *ggplot2* R package[57]. Second, genetic clusters were estimated using a

Bayesian clustering approach that incorporates Markov chain Monte Carlo (MCMC) simulations, as implemented in STRUCTURE v.2.3.4[27]. Run-length was set to 100,000 MCMC replicates after a burn-in period of 100,000 using correlated allele frequencies under a straight admixture model using the *locprior* option. We varied the number of clusters ($K$) from 1 to 17, with ten replicate runs for each value of $K$. The most likely number of clusters was determined using two methods including (1) plotting the log probability of the data $\ln \Pr(X \mid K)$ across the range of $K$ values tested and selecting $K$ where the value of $\ln \Pr(X \mid K)$ plateaued and the variance was minimized as recommended by the authors[27]; and 2) calculating $\Delta K$ values and employing the Evanno method[58]. DISTRUCT v.1.1 was used to visualize bar plots generated with the optimal number of $K$[59]. Finally, pairwise genetic differentiation for all island pairs was assessed using $\theta$[32], an unbiased measure of $F_{ST}$, and Analysis of Molecular Variance (AMOVA)[28] using GenoDive v.3.0[55]. Significance was tested using 10,000 permutations.

**Migration, kinship, and sex-biases.** Recent migration was estimated between islands using BayesAss[29]. This method uses multilocus genotypes and a Bayesian approach to generate posterior estimates of directional migration rates over the past several generations as implemented here in BA3-SNPs v.1.1[30]. Parameter tuning for MCMC acceptance was automated using ten runs of the BA3-SNPs-autotune python script and yielded the final mixing parameters for allele frequencies, inbreeding coefficients, and migration rates as 0.9991, 0.0875, and 0.9991, respectively. The final analysis included all individuals and was completed using 10 million iterations with one million iterations discarded as burn-in. Results were averaged across five replicate runs and migration rates were deemed significant if 95% credible sets ($\mu \pm \sigma \times 1.96$) did not contain zero following author recommendations[29].

Kinship coefficients were calculated between all individuals using the relative probability of allelic identity-by-descent[31], employed in GenoDive v.3.0[55]. We selected this estimator[31] because it makes no assumptions regarding Wright's inbreeding coefficient and is robust for estimating spatial genetic structure using codominant molecular markers characterized by low polymorphism[60]. Given that first- and second-order relatives are expected to have kinship coefficients of 0.250 and 0.125, respectively, we considered kinship coefficients $\geq 0.20$ and $<0.20$ and $\geq 0.10$ to be indicative of first- and second-order relatives, respectively, as genotyping error and missing data can decrease accuracy. Contemporary dispersal was inferred if first- or second-order relatives were detected on different islands.

Tests for sex-biased dispersal were performed using individuals aged $\geq 3$ in accordance with observations of mule deer dispersal[35,36], as sex-biased dispersal tests should only include mature individuals that have already dispersed. Initially, we calculated pairwise $\theta$[32] separately by sex between all islands where $n \geq 2$ for both males and females; in these comparisons, the dispersing sex should have lower $\theta$ values on average than the philopatric sex. Next, we calculated the mean (mAIC) and variance (vAIC) of population assignment indices separately for each sex ($n = 123$; 71 males and 52 females) using the "hierfstat" R package[33]. For each individual, the population assignment index is an estimation of the probability of each genotype arising from the sampled population; thus, the dispersing sex is expected to have a lower mean assignment index than the philopatric sex. Conversely, the dispersing sex is expected to have a greater variance of the assignment index because sampled individuals may be either residents or immigrants, while sampled individuals from the philopatric sex are more likely to be residents only. Given the high degree of connectivity found in the system, indices were calculated considering all individuals, except those from SGang Gwaay, as one population (see "Results").

**Statistics and reproducibility.** All RADseq libraries included technical replicates within and among libraries, which demonstrated low genotyping error and the reproducibility of recovered genotypes. All raw and genotypic data are accessioned (see "Data Availability"). Statistical analyses were conducted using the cited packages and reproducibility can be achieved using the parameters reported in the Methods.

**Reporting summary.** Further information on research design is available in the Nature Research Reporting Summary linked to this article.

## Data availability

All Illumina raw reads are available from the NCBI sequence read archive (BioProject ID: PRJNA803424); RAD tag sequences and SNP genotypic data are deposited in the Dryad Digital Repository (https://doi.org/10.5061/dryad.q2bvq83mq)[61].

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

## Acknowledgements

Haawa, haawa, haawa to all the members of the Gwaii Haanas field unit and Council of the Haida Nation who supported this work. Thanks/Merci as well to Jean-Louis Martin and his past students, and Keith Moore for helping with historical samples and discussion of cull history on Haida Gwaii. The Llgaay gwii sdiihlda project was funded through Parks Canada Agency's Conservation and Restoration (CoRe) Program and was led in technical aspects by Chris Gill of Coastal Conservation. Computational resources were made available by Compute Canada through the Resources for Research Group's program to M.A.R. Figures were created in part using BioRender.com. This study was funded by Parks Canada contribution agreement # GC-1031 and NSERC Discovery grant # RGPIN-2019-04621 (M.A.R.). Thank you to the editor and three anonymous reviewers who all made valuable suggestions that greatly improved the manuscript.

## Author contributions

B.T.B. participated in study design, conducted all laboratory work and data analyses, and drafted the manuscript. R.L.I. participated in study design, coordinated all field aspects of the study, and helped draft the manuscript. M.A.R. conceived, designed and coordinated the study, and helped draft the manuscript.

## Competing interests

The authors declare no competing interests.
