## [Transparent Peer Review File · Communications Biology]

Population genomics of Sitka black-tailed deer supports invasive species management and ecological restoration on islandsReviewers' comments:

Reviewer #1 (Remarks to the Author):

This paper analyses an interesting invasive system using genome-wide sequence data derived from RADseq. Overall it's important work that's very well written and clear - I enjoyed reading this manuscript and look forward to it being published.

That being said, at the moment there are several areas of the analysis that are either unclear or subject to bias, making it difficult to eliminate alternative interpretations of the results that would be at odds with the provided interpretations. These issues mostly stem from either the large differences in sample size among populations, or from the fact that the invasion is very recent.

These are:

1) The migration rate estimates are difficult to interpret as the invasion is only ~130 years old (how many generations is this, roughly? 30-40?). If there had been no migration at all following invasion of all the islands, what would the patterns look like? Would these populations still be quite similar genetically? Unless there's some way to deal with the high recent coancestry in this system, I do not think these migration rate analyses are appropriate.

It also appears that there is a relationship between significant estimates and sample size, where the migration rates and significance is higher for demes with larger n. Might the differential sample size be producing these results? Given that Moresby has the largest n and the highest estimated migration rates, this issue should be investigated. How do the results look if sample sizes are equalised?

2) Filtering for heterozygosity should follow different protocols than for structure. A recent paper showed that heterozygosity derived from SNPs was biased in various ways, but that heterozygosity was not biased when calculated from monomorphic and polymorphic sites (Schmidt et al 2021 Meth Eco Evo). This unbiased heterozygosity should be easy to recalculate as the Stacks Populations program outputs these results alongside the results from polymorphic sites only. At present, the heterozygosity estimates in this paper are biased by differential sample sizes among populations, by the minor allele frequency cut-offs applied, and by the inclusion of sites with missing genotypes.

3) For the sex-biased dispersal analysis, F_{st} -like indices can be biased by sample size, where smaller sample sizes give higher estimates. What do the results look like if the comparisons have equalised sample size? i.e. if each island comparison is limited to ensure $n(\text{males}) = n(\text{females})$.

The first of the above issues is the most important as this is what the conclusions rest upon. I think that the authors need to acknowledge these limitations of the recentness of the invasion (see Fitzpatrick et al 2012 Biol Invasions). The high differentiation at SGang Gwaay could be one means of getting around this – if the differentiation there could be shown to be due to isolation and not only due to the extreme bottleneck this population has undergone. One way to look at this could be through tree-based inferences such as in the program FineRADstructure (Malinsky et al 2018).

Also, the kinship analysis should not be affected by recent coancestry, and I think this element of the analysis could be presented more prominently through a figure showing the island pairs where kin have been observed. Several recent papers have used close kin to infer movement patterns in systems with high coancestry due to recent change.

Specific comments:

In the Introduction, I think it's critically important to mention when the invasion started and how many generations it's been since. It's hard to make sense of the results without considering the number of generations.

L167 Heterozygosity filtering. See above

L208 Is there a reference for why genotyping error and missing data would underestimate this, rather than overestimate or simply decrease general accuracy?

L 237 The PCA is very hard to read. I'd suggest using different shapes as well as colours, and/or using transparent filled shapes, as currently it's hard to tell which symbols are from which islands. Also, these figures should indicate the % of explained variance of each axis

L248 In the AMOVA, what do the P-values actually represent here, in terms of genetic structure? If there is no hierarchical structuring in the AMOVA outside of 'within' or 'among' islands, is it possible to get a non-significant P-value? Also, the P-values should either specify exact values or use the '<' sign rather than '='.

L267 It would be a good idea to show which island pairs had close kin on them, as this is referred to in the discussion but can't really be read from the Figure.

L273 Please report the sample sizes for this analysis. Also, Fst-like indices can be biased by sample size, where smaller sample size gives higher estimates. What do the results look like if the comparisons have normalised sample size? i.e. limit each island comparison to ensure $n(\text{males}) = n(\text{females})$.

L299 Considering that the invasion is so recent, how much differentiation would be expected? To make this claim it's necessary to put this into some context – what other recent invasion examples can be pointed to in order to indicate that the low differentiation here is due to ongoing connectivity and not due to recent coancestry?

Clearly the differentiation of SGang Gwaay is an interesting point around this, but of course this population has been extensively culled. So it is a possible alternative explanation that the differentiation of this island is simply due to culling?

Overall, it will be necessary to make the case that low differentiation reflects ongoing processes rather than historical contingencies, otherwise one will not be able to conclude much about current movement patterns.

L305 Much of this information on SGang Gwaay could be better placed in the introduction, so that the reader can have some context into why this population is so different.

L353 Though this is conditional on analyses that use equal sample sizes (see above comment), wasn't there a signal of female-biased dispersal? Why is this not discussed here?

Reviewer #2 (Remarks to the Author):

This well written MS presents data from a population genetic study of invasive deer (Sitka black-tailed) in the Haida Gwaii archipelago. The results could help management strategies of this invasive pest. The study used well established methods and have to my knowledge applied the necessary checks (error rates, variant filtering etc) and statistical analyses that are required for their study.

1. Please include GPS coordinates of the locations in the MS. This information can be added to Table 1 or included in a supplementary table. This makes the data more reusable and comparable to future studies. I think it would be helpful to also have the linear distance of each site from Moresby, again this could be included in the Table 1 or included in a supplementary table.
2. As F statistics and PCA can be influenced by missingness I would like to see data on genotype missingness - per individual or per population. Just to ensure that the missingness is random with respect to population. This could be included in the supplementary information.
3. It is great to see the authors have used replicate samples to estimate genotyping error, but these error rates seem fairly high (to me - if I assume a 'normal' range is between 0.1-5%. Perhaps my idea of acceptable error is a little low. I would suggest the authors report as "moderate" not "low".
4. Line 148 - Revise "Populations module" to "STACKS Population module" just to make it clearer.
5. Line 71 - Wording is a little awkward – can an attempt be incomplete? I think the eradication can be unsuccessful (i.e. deer are still present).
6. In Figure 1b and c. It is not exactly clear where on the Haida Gwaii each larger figure comes from. Maybe add a star or something to indicate where each of the larger 'zoomed-in' maps are from exactly.
7. Line 369. Change "demonstrated to be" to "have been"

Reviewer #3 (Remarks to the Author):

The manuscript of Burgess and co-authors uses a large sampling of the invasive Sitka black-tailed deer across 15 islands the Haida Gwaii archipelago. Using genomics and population genetics they efficiently investigate connectivity and gene flow of invasive deer and conclude that attempts to eradicate the species in these islands are doomed to fail.

The paper is well written, clear, and relevant to the readership of Communications Biology. The methods are adequate and well presented. The results are clearly presented and well discussed. Altogether the manuscript represents a great contribution to science and to species management and ecological restoration on islands. It deserves to be published in Communications Biology

The manuscript does not require major changes. However, the authors should make an effort to efficiently represent their results. I have made a couple of suggestions to that regard, below, that will improve the ability of the reader to interpret comprehensively their results.

I also suggest an important clarification on the way the genotyping thresholds were chosen, which is believe is important to clarify.

I have made a few final comments on the structure of the discussion which could be shortened and re-centered to focus exclusively on the results and avoid repetitions with the introduction.

Introduction:

The use of the term genotyping by sequencing (GBS) is confusing, since it also refers to a particular RAD approach. Make sur to use RADseq consistently across the whole ms.

L94 : remove « large » : displayed a signature of genetic

M&M

L148-158: why using “quality and quantity of retained individuals and loci by comparing them to the amount of missing data per individual and average depth of coverage per locus” when you could assess “genotyping error as the discordance of genotype calls between replicate samples using a custom python script (<https://github.com/bsjodin/genoerrorcalc>)” as mentioned above (L124). Error rate is at least as important as depth and missing data, if not way more relevant at this stage.

L159: describe, explain, clarify what you are calling an “outlier”

Results

L223: The criteria to select these filtering parameters were not crystal clear in the M&M and the results at line 223 add no clarification.

L227: What about genotyping error for other population parameters set?

L257: replace “several” by the exact number of pairs

L268-269: represent all first order relative geographically (within and across islands), eventually on the same map that will represent exhaustively all island-island migrations inferred. Eventually represent island-island 2nd order relative too.

L275: reformulate mAIC sentence to emphasize that it is a different approach giving a slightly different result.

Discussion

L281-292: Does your discussion needs an introduction just repeating the introduction of the ms? Or does it need a wrap-up of the unique ecological and molecular and set-up that allows to

L293: replace “genotyping by sequencing” by “RADseq”

L325-327: clarify this last sentence, and remind that the estimated differentiation is a function of connectivity and of population size

Managment implication

L362-374: Does your Managment implication section needs and introduction? It should be tightly linked to the results and start straight with these results implications. All other fact can appear in the following text as a discussion of your results.

L411 423: Favor a more careful discussion here, considering the island size, the maximum distance crossed and the migration rates across the study, how likely is it that in SGang Gwaay, the results exhibit a temporary pattern and that migration will occur in the near future?

Figures and tables.

Figure 2ab: Indicate the percentage of variance explained by each of your axes on the pca.

Figure 2ab: Use different symbols, on top of more distinct colors, to facilitate interpretation of the pca.

Figure 2cd: Aside the barplot representation, represent structure results geographically (i.e. on a map) to facilitate the result interpretation.

Table S2: A similar table should be presented to get along the subsample analysis (Fig2d). If possible consider turning this table into a set of plot that is more comfortable to interpret.

Table S3: Why tableS3 has 9 signification values but their quantitative representation (Fig 1bc) has only 5 arrows. This is the most relevant results as stated the title and abstract and should accurately represented.

Reviewers' comments:

Reviewer #1 (Remarks to the Author):

This paper analyses an interesting invasive system using genome-wide sequence data derived from RADseq. Overall it's important work that's very well written and clear - I enjoyed reading this manuscript and look forward to it being published.

That being said, at the moment there are several areas of the analysis that are either unclear or subject to bias, making it difficult to eliminate alternative interpretations of the results that would be at odds with the provided interpretations. These issues mostly stem from either the large differences in sample size among populations, or from the fact that the invasion is very recent.

These are:

1) The migration rate estimates are difficult to interpret as the invasion is only ~130 years old (how many generations is this, roughly? 30-40?). If there had been no migration at all following invasion of all the islands, what would the patterns look like? Would these populations still be quite similar genetically? Unless there's some way to deal with the high recent coancestry in this system, I do not think these migration rate analyses are appropriate.

It also appears that there is a relationship between significant estimates and sample size, where the migration rates and significance is higher for demes with larger n. Might the differential sample size be producing these results? Given that Moresby has the largest n and the highest estimated migration rates, this issue should be investigated. How do the results look if sample sizes are equalised?

RESPONSE: Thank you for this thoughtful feedback. The generation time for Sitka black-tailed deer has been estimated at 1.5-2.5 years. Given this value, we estimated 50-80 generations have passed since the initial invasion in the 1890s and now provide these details in the introduction (LINES 56-57). The recentness of the invasion led us to investigate connectivity using several different approaches, including Weir and Cockerham's θ (Weir and Cockerham, 1984), which is an unbiased measure of F_{ST} designed to accommodate, among other things, unequal sample sizes (LINES 194-195). We also specifically chose the approach of Wilson and Rannala (2003) implemented in BAYESASS to investigate contemporary migration, which is designed to accommodate shared polymorphism due to recent ancestry in estimates of recent migration (i.e. over the last several generations; Wilson and Rannala, 2003); we now highlight this timeframe explicitly (LINES 200-203). The findings from Weir and Cockerham's θ and BAYESASS were reinforced by the kinship analyses, which represent the most direct and finest scale approach for detecting contemporary movement in our study; in this case, we detected several instances of first- and second-order related individuals across islands within Juan Perez Sound, which is now highlighted in new Figure 3, and provides additional confidence in our overall findings.

2) Filtering for heterozygosity should follow different protocols than for structure. A recent paper showed that heterozygosity derived from SNPs was biased in various ways, but that heterozygosity was not biased when calculated from monomorphic and polymorphic sites (Schmidt et al 2021 Meth Eco Evo). This unbiased heterozygosity should be easy to recalculate

as the Stacks Populations program outputs these results alongside the results from polymorphic sites only. At present, the heterozygosity estimates in this paper are biased by differential sample sizes among populations, by the minor allele frequency cut-offs applied, and by the inclusion of sites with missing genotypes.

RESPONSE: As per your suggestion, we now also include estimates of observed and expected heterozygosity using all sites, which are summarized in the Results (LINES 249-251) and reported in a revised Table 1.

3) For the sex-biased dispersal analysis, Fst-like indices can be biased by sample size, where smaller sample sizes give higher estimates. What do the results look like if the comparisons have equalised sample size? i.e. if each island comparison is limited to ensure $n(\text{males}) = n(\text{females})$.

RESPONSE: This is a good point, but note we explicitly tried to accommodate these considerations by taking into account the biology of the species (only using individuals aged ≥ 3 in accordance with observations of mule deer dispersal) and a minimal sample size for an island to be included ($n \geq 2$ for both males and females). As you can suspect, sample sizes per appropriately aged individual per sex quickly reach small sizes. We also attempted to deal with this concern by using an unbiased measure of Fst (θ ; Weir and Cockerham, 1984) designed to accommodate, among other things, unequal sample sizes. All that said, we still think it worth including the results of these analyses, but we have now more explicitly note the limitations in the Discussion. (LINES 378-382)

The first of the above issues is the most important as this is what the conclusions rest upon. I think that the authors need to acknowledge these limitations of the recentness of the invasion (see Fitzpatrick et al 2012 Biol Invasions). The high differentiation at SGang Gwaay could be one means of getting around this – if the differentiation there could be shown to be due to isolation and not only due to the extreme bottleneck this population has undergone. One way to look at this could be through tree-based inferences such as in the program FineRADstructure (Malinsky et al 2018).

RESPONSE: Please see our response to the first point raised above. Given the diverse analyses already implemented and the congruent signal obtained, we did not explore FineRADstructure (Malinsky et al 2018) as an additional approach. If this is considered a significant omission, we can look to explore this approach if required.

Also, the kinship analysis should not be affected by recent coancestry, and I think this element of the analysis could be presented more prominently through a figure showing the island pairs where kin have been observed. Several recent papers have used close kin to infer movement patterns in systems with high coancestry due to recent change.

RESPONSE: Based on this feedback, we prepared a new Figure 3 to more prominently present the results of the kinship analyses.

Specific comments:

In the Introduction, I think it's critically important to mention when the invasion started and how many generations it's been since. It's hard to make sense of the results without considering the number of generations.

RESPONSE: Done. (LINES 56-57)

L167 Heterozygosity filtering. See above

RESPONSE: As per your suggestion, we now also include estimates of observed and expected heterozygosity using all sites, which are summarized in the Results (LINES 249-251) and reported in a revised Table 1.

L208 Is there a reference for why genotyping error and missing data would underestimate this, rather than overestimate or simply decrease general accuracy?

RESPONSE: We revised “cause underestimations” to “decrease accuracy”. (LINE 218)

L 237 The PCA is very hard to read. I'd suggest using different shapes as well as colours, and/or using transparent filled shapes, as currently it's hard to tell which symbols are from which islands. Also, these figures should indicate the % of explained variance of each axis

RESPONSE: We revised both PCAs in Figure 2. Transparent filled shapes were added to make it easier to distinguish islands. The percentage of explained variance was added to each axis and all labels were made larger.

L248 In the AMOVA, what do the P-values actually represent here, in terms of genetic structure? If there is no hierarchical structuring in the AMOVA outside of 'within' or 'among' islands, is it possible to get a non-significant P-value? Also, the P-values should either specify exact values or use the '<' sign rather than '='.

RESPONSE: This is a good point; given the number of loci used, it is highly unlikely to not obtain a significant result; the value of this analysis lies in the relative values which we focused on in the reporting of the results (LINES 267-271). We have retained the *p*-values for completeness and edited the sign to “<” throughout the text.

L267 It would be a good idea to show which island pairs had close kin on them, as this is referred to in the discussion but can't really be read from the Figure.

RESPONSE: As recommended above, a new figure (Fig.3) has been added to the manuscript, showing the geographic relationships between all first and second order relatives.

L273 Please report the sample sizes for this analysis. Also, F_{st} -like indices can be biased by sample size, where smaller sample size gives higher estimates. What do the results look like if the comparisons have normalised sample size? i.e. limit each island comparison to ensure $n(\text{males}) = n(\text{females})$.

RESPONSE: Please see our response above. The sample size for the mAIC and vAIC analyses are now explicitly indicated in the methods (i.e. “(n=123; 71 males and 52 females)”) to improve clarity. (LINE 226)

L299 Considering that the invasion is so recent, how much differentiation would be expected? To make this claim it's necessary to put this into some context – what other recent invasion examples can be pointed to in order to indicate that the low differentiation here is due to ongoing connectivity and not due to recent coancestry?

Clearly the differentiation of SGang Gwaay is an interesting point around this, but of course this population has been extensively culled. So it is a possible alternative explanation that the differentiation of this island is simply due to culling?

Overall, it will be necessary to make the case that low differentiation reflects ongoing processes rather than historical contingencies, otherwise one will not be able to conclude much about current movement patterns.

RESPONSE: Please see response above. We used a variety of approaches to attempt to disentangle shared polymorphism due to recent ancestry and sample size effects from estimates of differentiation and recent migration, including the use of an unbiased measure of F_{st} (θ ; Weir and Cockerham, 1984), an analysis developed to estimate contemporary migration rates over the past several generations (Wilson and Rannala, 2003) and kinship analyses that can more directly detect evidence of recent gene flow. We have attempted to emphasize this a bit more in the Discussion (LINES 322-325). Lastly, we make an explicit point to discuss the potential impacts of management interventions on observed patterns of within-island variation and among-island differentiation, especially as they relate to SGang Gwaay (LINES 343-346).

L305 Much of this information on SGang Gwaay could be better placed in the introduction, so that the reader can have some context into why this population is so different.

RESPONSE: Done. (LINES 79-84)

L353 Though this is conditional on analyses that use equal sample sizes (see above comment), wasn't there a signal of female-biased dispersal? Why is this not discussed here?

RESPONSE: The only analysis that supported a signal of female-biased dispersal was vAIC, which was not statistically significant. The only significant test for sex-biased dispersal was the pairwise θ , which indicated that females were significantly more differentiated than males, suggesting male-biased dispersal. All that said, we have now more explicitly noted the limitations of these analyses in the Discussion per the previous comment/response above. (LINES 378-382)

Reviewer #2 (Remarks to the Author):

This well written MS presents data from a population genetic study of invasive deer (Sitka black-tailed) in the Haida Gwaii archipelago. The results could help management strategies of this invasive pest. The study used well established methods and have to my knowledge applied the necessary checks (error rates, variant filtering etc) and statistical analyses that are required for their study.

1. Please include GPS coordinates of the locations in the MS. This information can be added to Table 1 or included in a supplementary table. This makes the data more reusable and comparable to future studies. I think it would be helpful to also have the linear distance of each site from Moresby, again this could be included in the Table 1 or included in a supplementary table.

RESPONSE: Based on this feedback, we created a new supplementary table (Table S1) containing GPS coordinates for each island, as well as their linear distance from Moresby.

2. As F statistics and PCA can be influenced by missingness I would like to see data on genotype missingness - per individual or per population. Just to ensure that the missingness is random with respect to population. This could be included in the supplementary information.

RESPONSE: We added mean missing genotypic data percentages by population to new Table S1.

3. It is great to see the authors have used replicate samples to estimate genotyping error, but these error rates seem fairly high (to me - if I assume a 'normal' range is between 0.1-5%. Perhaps my idea of acceptable error is a little low. I would suggest the authors report as "moderate" not "low".

RESPONSE: Given that terms like "low" and "moderate" can be subjective, we removed the adjective ("low") from this sentence. (LINE 243)

4. Line 148 – Revise "Populations module" to "STACKS Population module" just to make it clearer.

RESPONSE: Done. (LINE 155)

5. *Line 71 - Wording is a little awkward – can an attempt be incomplete? I think the eradication can be unsuccessful (i.e. deer are still present).*

RESPONSE: Although a bit awkward, ‘incomplete’ is the correct classification for such eradications as defined in the Database of Island Invasive Species Eradications (DIISE); we have added this reference for clarity (LINE 73).

6. *In Figure 1b and c. It is not exactly clear where on the Haida Gwaii each larger figure comes from. Maybe add a star or something to indicate where each of the larger ‘zoomed-in’ maps are from exactly.*

RESPONSE: We revised the legend to better provide this context. (LINE 686-687)

7. *Line 369. Change “demonstrated to be” to “have been”*

RESPONSE: Done. (LINE 393)

Reviewer #3 (Remarks to the Author):

The manuscript of Burgess and co-authors uses a large sampling of the invasive Sitka black-tailed deer across 15 islands the Haida Gwaii archipelago. Using genomics and population genetics they efficiently investigate connectivity and gene flow of invasive deer and conclude that attempts to eradicate the species in these islands are doomed to fail.

The paper is well written, clear, and relevant to the readership of Communications Biology. The methods are adequate and well presented. The results are clearly presented and well discussed. Altogether the manuscript represents a great contribution to science and to species management and ecological restoration on islands. It deserves to be published in Communications Biology. The manuscript does not require major changes. However, the authors should make an effort to efficiently represent their results. I have made a couple of suggestions to that regard, below, that will improve the ability of the reader to interpret comprehensively their results.

I also suggest an important clarification on the way the genotyping thresholds were chosen, which is believe is important to clarify.

I have made a few final comments on the structure of the discussion which could be shortened and re-centered to focus exclusively on the results and avoid repetitions with the introduction.

Introduction:

The use of the term genotyping by sequencing (GBS) is confusing, since it also refers to a particular RAD approach. Make sur to use RADseq consistently across the whole ms.

RESPONSE: We changed “genotyping-by-sequencing” to “restriction-site associated DNA sequencing” throughout the text.

L94 : remove « large » : displayed a signature of genetic

RESPONSE: We changed “displayed a large signature of genetic” to “displayed a signature of genetic”. (LINE 101)

M&M

L148-158: why using “quality and quantity of retained individuals and loci by comparing them to the amount of missing data per individual and average depth of coverage per locus” when you could assess “genotyping error as the discordance of genotype calls between replicate samples using a custom python script (<https://github.com/bsjodin/genoerrorcalc>)” as mentioned above (L124). Error rate is at least as important as depth and missing data, if not way more relevant at this stage.

RESPONSE: We calculated genotyping error for the extreme parameter sets and added them to Table S2 to demonstrate the extent to which these values vary across the different options (including the one ultimately chosen). We now also include this consideration in the methods. (LINES 162-164)

L159: describe, explain, clarify what you are calling an “outlier”

RESPONSE: Done. We edited the sentence to improve clarity: “Loci were considered outliers and removed if they had a mean q-value <0.20 averaged over five runs”. (LINES 170-171)

Results

L223: The criteria to select these filtering parameters were not crystal clear in the M&M and the results at line 223 add no clarification.

RESPONSE: We added the following sentence to the methods to be more explicit: “The final parameters were selected to maximize the number of loci, while maintaining low missing data, high mean depth of coverage and reasonable within/among library genotyping error rates.”. (LINES 162-164)

L227: What about genotyping error for other population parameters set?

RESPONSE: We calculated genotyping error for the extreme parameter sets and added them to Table S2 to demonstrate the extent to which these values vary across the different options (including the one ultimately chosen). We now also include this consideration in the methods. (LINES 162-164)

L257: replace “several” by the exact number of pairs

RESPONSE: We replaced “several” with “nine”. (LINE 276)

L268-269: represent all first order relative geographically (within and across islands), eventually on the same map that will represent exhaustively all island-island migrations inferred. Eventually represent island-island 2nd order relative too.

RESPONSE: Based on this feedback, we prepared a new Figure 3 to more prominently present the results of the kinship analyses.

L275: reformulate mAIC sentence to emphasize that it is a different approach giving a slightly different result.

RESPONSE: Done. We added the following sentences to improve clarity: “For each individual, the population assignment index is an estimation of the probability of each genotype arising from the sampled population; thus, the dispersing sex is expected to have a lower mean assignment index than the philopatric sex. Conversely, the dispersing sex is expected to have a greater variance of the assignment index because sampled individuals may be either residents or immigrants, while sampled individuals from the philopatric sex are more likely to be residents only.” (LINES 227-232)

Discussion

L281-292: Does your discussion needs an introduction just repeating the introduction of the ms? Or does it need a wrap-up of the unique ecological and molecular and set-up that allows to

RESPONSE: We recognize there are different approaches for beginning a Discussion. We prefer to use the first paragraph to contextualize what is to follow and have retained the wording as in the original.

L293: replace “genotyping by sequencing” by “RADseq”

RESPONSE: We revised to “population genomics”. (LINE 312)

L325-327: clarify this last sentence, and remind that the estimated differentiation is a function of connectivity and of population size

RESPONSE: Done. We revised this sentence to: “However, given that genetic differentiation is a function of both connectivity and population size, we cannot exclude the possibility that the

apparent lack of gene flow detected between SGang Gwaay and other islands is due to geographic isolation rather than a previous population bottleneck”. (LINES 343-346)

Management implication

L362-374: Does your Management implication section needs and introduction? It should be tightly linked to the results and start straight with these results implications. All other fact can appear in the following text as a discussion of your results.

RESPONSE: We feel that the opening paragraph of section provides a more detailed context to the subsequent discussion than what we are able to cover in the Introduction. We have elected to retain as in the original, but are certainly open to cutting if deemed necessary by the editor.

L411 423: Favor a more careful discussion here, considering the island size, the maximum distance crossed and the migration rates across the study, how likely is it that in SGang Gwaay, the results exhibit a temporary pattern and that migration will occur in the near future?

RESPONSE: Thank you for raising this point. We have added the following sentences to provide further reasoning supporting our conclusion that SGang Gwaay is a promising target for management: “For this study, deer were selectively harvested from the southern end of Gwaii Haanas, including twelve deer from the southern coast of Moresby, to best evaluate any possible connectivity with deer on SGang Gwaay. Given the heavy sampling that took place on SGang Gwaay (n=23), we expected that any gene flow to or from Moresby would be detected.” (LINES 435-439). In addition, we removed the word “strong” on LINE 447.

Figures and tables.

Figure 2ab: Indicate the percentage of variance explained by each of your axes on the pca.

RESPONSE: Done.

Figure 2ab: Use different symbols, on top of more distinct colors, to facilitate interpretation of the pca.

RESPONSE: We revised both PCAs in Figure 2 accordingly. Transparent filled shapes were added to make it easier to distinguish islands and all labels were made larger.

Figure 2cd: Aside the barplot representation, represent structure results geographically (i.e. on a map) to facilitate the result interpretation.

RESPONSE: Given the lack of structure detected throughout the system, we have elected not to add barplots to a map, especially given the addition of a new map figure (Figure 3) requested to more explicitly emphasize the geographical context of the kinship results.

Table S2: A similar table should be presented to get along the subsample analysis (Fig2d). If possible consider turning this table into a set of plot that is more comfortable to interpret.

RESPONSE: Done (Table S4).

Table S3: Why tableS3 has 9 significance values but their quantitative representation (Fig 1bc) has only 5 arrows. This is the most relevant results as stated the title and abstract and should accurately represented.

RESPONSE: Thank you for pointing this out. We modified Fig1b to include five arrows. Fig1c includes two arrows. The remaining two significant estimates of gene flow (Moresby -> Louise; Moresby -> Graham), are not shown, as these islands are not within Gwaii Haanas. We now explicitly indicate this in the Figure 1 legend. (LINES 685-686)

REVIEWERS' COMMENTS:

Reviewer #1 (Remarks to the Author):

The authors have done a good job of responding to my comments and I am happy to approve this manuscript